# Transcriptomic and Proteomic Insights into the Effect of Sterigmatocystin on *Aspergillus flavus*

**DOI:** 10.3390/jof9121193

**Published:** 2023-12-13

**Authors:** Yarong Zhao, Rui Zeng, Peirong Chen, Chulan Huang, Kaihang Xu, Xiaomei Huang, Xu Wang

**Affiliations:** 1Institute of Quality Standard and Monitoring Technology for Agro-Product of Guangdong Academy of Agricultural Sciences, Guangzhou 510640, China; zhaoyarong@gdaas.cn (Y.Z.); zengrui@gdaas.cn (R.Z.); chenpeirong@gdaas.cn (P.C.); huangchulan@gdaas.cn (C.H.); xukaihang@gdaas.cn (K.X.); huangxiaomei@gdaas.cn (X.H.); 2Laboratory of Quality and Safety Risk Assessment for Agro-Product (Guangzhou), Ministry of Agriculture, Guangzhou 510640, China

**Keywords:** *Aspergillus flavus*, aflatoxins, sterigmatocystin, transcriptome, proteomic

## Abstract

*Aspergillus flavus* is an important fungus that produces aflatoxins, among which aflatoxin B_1_ (AFB_1_) is the most toxic and contaminates food and poses a high risk to human health. AFB_1_ interacts with another mycotoxin sterigmatocystin (STC), which is also a precursor of AFB_1_. Herein, we determined the effect of STC on AFB_1_ by evaluating *A. flavus* transcriptomic and proteomic profiles in the presence or absence of STC by RNA-seq and isobaric tagging, respectively. Overall, 3377 differentially expressed genes were identified by RNA-seq. These genes were mainly associated with the cellular component organisation and biosynthesis, the synthesis of valine, leucine, and isoleucine, and the synthesis of aflatoxin. Clustered genes responsible for AFB_1_ biosynthesis exhibited varying degrees of downregulation, and *norB* expression was completely suppressed in the experimental group. During proteomic analysis, 331 genes were differentially expressed in response to STC. These differentially expressed proteins were associated with cell parts and catalytic and antioxidant activities. Differentially expressed proteins predominantly participated in metabolic pathways associated with aflatoxin biosynthesis, glycolysis/gluconeogenesis, glutathione metabolism, and carbon metabolism. Notably, the upregulated and downregulated enzymes in carbohydrate and glutathione metabolisms may serve as potential gateways for inhibiting aflatoxin biosynthesis. Moreover, twelve proteins including seven downregulated ones involved in aflatoxin biosynthesis were identified; among them, AflG was the most downregulated, suggesting that it may be the key enzyme responsible for inhibiting aflatoxin synthesis. These findings provide novel insights into *A. flavus* control and the mechanisms regulating mycotoxin production.

## 1. Introduction

Aflatoxin B_1_ (AFB_1_) is one of the most important mycotoxins worldwide, frequently detected at high concentrations as a contaminant in human and animal food [1,2,3]. It is a polyketide synthesised via secondary metabolic pathways in *Aspergillus*. Furthermore, it can cause pathophysiological changes in some organisms, including reduced growth rate, disturbance in the gastrointestinal tract, suppressed immune responses, and malnutrition. It also induces various histopathological manifestations in hepatocytes, such as the proliferation of the bile duct and fatty degeneration of hepatocytes [4,5,6,7]. In addition to its harmful effects in humans, AFB_1_ has been associated with developing hepatocellular carcinoma in various animals, including rodents, poultry, non-human primates, and fishes [8,9]. Therefore, the International Agency for Research on Cancer (IARC) classifies AFB_1_ as a group 1 compound, indicating that it is carcinogenic to humans [10].

Sterigmatocystin (STC) is another important mycotoxin primarily produced by *Aspergillus nidulans* and *Aspergillus versicolor*, although it can also be produced by other species belonging to the genera *Aschersonia*, *Botryotrichum*, *Emericella*, and *Chaetomium* [11,12]. STC is a potential carcinogen, mutagen, and teratogen and is categorised by the IARC as a group 2 compound [13]. Moreover, STC shares a biosynthetic pathway with AFB_1_ before STC synthesis, and the relative concentrations of AFB_1_ and STC in contaminated foods vary depending on the invasive species. *Aspergillus versicolor* and *A. nidulans* are unable to convert STC into O-methylSTC, which is a direct AFB_1_ precursor. Therefore, food infected with these fungi often have high STC concentrations. In contrast, when *A. flavus* and *A. parasiticus*, which are capable of metabolising STC, invade food, STC concentrations are generally low [11,14,15,16].

The simultaneous occurrence of multiple mycotoxins in a single product is common, and the co-occurrence of AFB_1_ and STC has been reported in food products [17,18,19,20]. Exposure to multiple mycotoxins can cause synergistic, additive, or antagonistic toxic effects, and predicting the toxicity of combined mycotoxins solely based on individual toxicity is challenging [21,22,23,24]. Mycotoxins that share modes of action are generally expected to have the least additive effects [25]. AFB_1_ exerts toxicity via DNA modification, metabolic activation, cell death/transformation, and cell deregulation [26,27,28,29]. In contrast, STC exerts its toxicity through DNA adduct formation, inhibition of the cell cycle and mitosis, and the promotion of ROS formation and lipid peroxidation [30,31,32,33]. Thus, there are similarities as well as differences between the modes of actions of AFB_1_ and STC.

To our knowledge, two studies have shown that precursors in the aflatoxin biosynthesis pathway could be used as potential early warning markers for aflatoxin contamination in these strains that produce aflatoxins [34,35]. However, there is a poor understanding of the interactions between STC and AFB_1_ in food contamination.

Therefore, in this study, we aimed to investigate differentially expressed genes (DEGs) and differentially expressed proteins (DEPs) in *A. flavus* after introducing STC into the matrix using a combination of transcriptomic and proteomic data. The findings of this study offer new insights into the control of *A. flavus* and the mechanism that regulates mycotoxin production, and that will be useful for the prevention and control of food poisoning.

## 2. Materials and Methods

### 2.1. Strains and Sample Preparation

The aflatoxigenic isolate *A. flavus* (CGMCC 3.2980) was purchased from the China General Microbiological Culture Collection Center and maintained on potato dextrose agar (PDA) slants at 4 °C until activation. *Aspergillus flavus* was cultured on PDA for 7 d at 30 °C; thereafter, conidia was harvested aseptically and suspended homogeneously in sterile distilled water containing 0.05% Tween 80. Conidial concentration was determined using a haemocytometer and adjusted to 1 × 10^7^ conidia mL^−1^ before use in the experiments.

### 2.2. Inoculation

A hundred millilitres of potato dextrose broth (PDB) medium was spiked with 10 µg/mL STC and inoculated with 1 mL of the conidial suspension. Three replicate flasks were prepared, labelled as TJ-1, TJ-2, and TJ-3, while the control flasks that contained only A. flavus conidia without STC were labelled as CK-1, CK-2, and CK-3. The flasks were incubated at 30 °C in darkness for 5 d.

### 2.3. Transcriptomic Analysis of A. flavus

#### 2.3.1. Total RNA Extraction and Quality Testing

Total RNA was extracted from cultured PDB media containing 10 µg/mL STC using TRIzol^®^ reagent according to the instructions of the manufacturer (Invitrogen, Stoney Creek, CA, USA). Genomic DNA was removed using DNase I (TaKara, Kyoto, Japan). The quality of the extracted RNA was determined using a 2100 Bioanalyzer (Agilent, Stoney Creek, CA, USA) and quantified using an ND-2000 (NanoDrop Technologies, Wilmington, DE, USA).

#### 2.3.2. RNA-Seq Library Construction and Sequencing

An RNA-seq transcriptome library was prepared using the TruSeq^TM^ RNA Sample Preparation Kit (Illumina, San Diego, CA, USA) with 5 μg of total RNA. mRNA was isolated using oligo (dT) beads through the polyA selection method and fragmented using a fragmentation buffer. Double-stranded cDNA was synthesised using a SuperScript double-stranded cDNA synthesis kit (Invitrogen, Carlsbad, CA, USA) with random hexamer primers (Illumina). The synthesised cDNA was subjected to end-repair, phosphorylation, and ‘A’ base addition according to the library construction protocol of Illumina. After selecting the RNA-seq libraries, cDNA target fragments ranging from 200 to 300 bp were selected using a 2% low-range ultra-agarose gel, followed by PCR amplification using Phusion DNA polymerase (NEB) with 15 PCR cycles. After quantification using TBS380, the paired-end RNA-seq library was sequenced using Illumina HiSeq Xten (2 × 150 bp read length).

#### 2.3.3. Mapping Reads and Sequence Assembly

Raw paired-end reads were subjected to trimming and quality control using SeqPrep (https://github.com/jstjohn/SeqPrep, accessed on 19 March 2019) and Sickle (https://github.com/najoshi/sickle (accessed on 19 March 2019), version 1.2) with default parameters. The resulting clean reads were separately aligned to reference genomes using TopHat orientation mode software 2.1.1 (http://tophat.cbcb.umd.edu (accessed on 19 March 2019), version 2.1.1). The mapping criteria of TopHat for the alignment were set to allow sequencing reads to uniquely match the genome, with a maximum of two mismatches, without insertions or deletions. The gene regions were expanded according to the depths of the sites and operons obtained. In addition, the whole genome was divided into multiple 15 kbp windows that shared 5 kbp. Newly transcribed regions were defined as those with more than two consecutive windows without overlapping regions of the gene, with a minimum requirement of two mapped reads per window in the same orientation.

#### 2.3.4. Differential Expression Analysis and Functional Enrichment

To identify the DEGs among the paired samples, the expression level of each transcript was calculated according to the fragments per kilobase of exon per million mapped reads method (FRKM). RSEM (http://www.biomedsearch.com/nih/RSEM-accurate-transcript-quantification-from/21816040.htm, accessed on 19 March 2019) was used to quantify gene abundance. The R statistical package software EdgeR 3. 14. 0 (Empirical Analysis of Digital Gene Expression in R [http://www.bioconductor.org/packages/2.12/bioc/html/edgeR.html (accessed on 19 March 2019), version 3.14.0]) was used for differential expression analysis. In addition, functional enrichment analysis, including enrichment analysis of GO terms and KEGG pathways, was conducted to identify the DEGs and metabolic pathways that exhibited significant enrichment of GO terms compared to the whole-transcriptome background (using the expression difference fold change and FDR statistical methods to determine significance; the screening criteria is FDR < 0.05, |log_2_FC| ≥ 1). GO functional enrichment and KEGG pathway analyses were performed using the Goatools (https://github.com/tanghaibao/Goatools (accessed on 19 March 2019), version 0.6.5) and KOBAS (http://kobas.cbi.pku.edu.cn/home.do (accessed on 19 March 2019), version 2.1.1) databases, respectively.

### 2.4. Proteomic Analysis of A. flavus

#### 2.4.1. Total Protein Extraction

Briefly, total protein extraction from *A. flavus* was performed using urea lysis buffer (7 M urea, 2 M thiourea, and 1% SDS) supplemented with a protease inhibitor. The protein concentrations were detected using a BCA Protein Assay Kit (Pierce, Thermo, Waltham, MA, USA). Following reduction, cysteine alkylation, and digestion, the samples were labelled with the isobaric tagging for relative and absolute quantitation (iTRAQ) reagent (Applied Biosystems, Waltham, MA, USA, 4390812) according to the manufacturer’s instructions. The peptides were then desalted using C18 solid-phase extraction and subjected to nanoliquid chromatography–mass spectrometry/mass spectrometry (LC-MS/MS) analysis.

#### 2.4.2. Protein Digestion and iTRAQ Labelling

Protein digestion was performed following a standard procedure, and the resulting peptide mixture was labelled using an 8-plex iTRAQ reagent (Thermo Fisher, Waltham, MA, USA, No. 90111) according to the instruction of the manufacturer. Briefly, 100 μg of total protein from each sample was mixed with 100 μL of lysate. TCEP (10 mM) was added to this mixture and incubated at 37 °C. After 60 min, iodoacetamide (40 mM) was added, and the mixture was stored in the dark at 25 °C for 40 min.

Six-fold volume of cold acetone was added at −20 °C to precipitate protein for 4 h. Following centrifugation at 10,000× *g* at 4 °C for 20 min, the pellet was re-suspended in 100 µL of 50 mM triethylammonium bicarbonate (TEAB) buffer. Trypsin was added at a 1:50 trypsin-to-protein mass ratio and incubated at 37 °C overnight. One unit of iTRAQ reagent was thawed and reconstituted in 50 µL of acetonitrile. After tagging for 2 h at room temperature, hydroxylamine was added and allowed to react for 15 min at room temperature.

#### 2.4.3. RPLC Separation

The pooled samples were fractionated into fractions using ACQUITY Ultra Performance liquid chromatography (Waters, Milford, MA, USA) with an ACQUITY UPLC BEH C18 Column (1.7 µm, 2.1 mm × 150 mm, Waters, USA) to enhance proteomic depth. Briefly, peptides were first separated with a gradient elution mobile phase (Phase B: 5 mM Ammonium hydroxide solution containing 80% acetonitrile, pH 10) over 48 min at a 200 μL/min flow rate. From each sample, 20 fractions were collected, and they were subsequently pooled to obtain ten fractions per sample.

#### 2.4.4. LC-MS/MS Analysis

The labelled peptides were subjected to online nano-flow liquid chromatography-tandem mass spectrometry using a 9RKFSG2_NCS-3500R system (Thermo, Waltham, MA, USA) connected to a Q Exactive Plus quadrupole Orbitrap mass spectrometer (Thermo, Waltham, MA, USA) via a nanoelectrospray ion source. The Q Exactive Plus instrument was operated in the data-dependent acquisition mode (DDA), allowing an automatic switch between full-scan MS and MS/MS acquisition.

#### 2.4.5. Protein Identification and Annotation

The RAW data files were analysed using Proteome Discoverer (Thermo Scientific, Waltham, MA, USA, Version 2.2) against *A. flavus* database (ftp://ftp.ensemblgenomes.org/pub/fungi/release-38/fasta/aspergillus_flavus/pep/, accessed on 7 December 2017). Table 1 shows the search criteria used for the MS/MS. Thresholds of fold change > 1.5 and *p*-value < 0.05 were used to identify DEPs. All identified proteins were annotated using the GO (http://www.blast2go.com/b2ghome; http://geneontology.org/, accessed on 15 November 2023) and KEGG (http://www.genome.jp/kegg/, accessed on 1 November 2023) pathways. The DEPs were then used for GO and KEGG enrichment analyses.

## 3. Results

### 3.1. Transcriptomic Analysis of A. flavus Cultured in PDB with STC

#### 3.1.1. Summary of RNA-Seq Data Sets

Transcriptome sequencing of the six samples (two groups, three duplicates) of *A. flavus* CGMCC 3.2980 generated 30.31 million raw reads. Among the total read pairs, 29.38 million passed the quality standards. Base composition and mass analysis revealed that there were few low-quality bases, meeting the sequencing quality control standard and the requirements for subsequent bioinformatic analyses.

Comparative analysis of the *A. flavus* genomes was performed using the clean reads after filtering with TopHat 2.0. The analysis showed that >85% of the reads mapped uniquely to the *A. flavus* genome. Furthermore, the proportion of multiple mapped reads or fragments that coincided with a standard half-point ratio was below 10% in multiple localisation sequencing, indicating that the reference genome was appropriate and the three were not contaminated during the experiment.

Gene expression levels depend on transcriptional abundance, where lower transcriptional abundance indicates lower gene expression levels. In this study, the overall transcriptional activity of genes was quantified by calculating the number of reads per kilobase of exon per million mapped reads (FPKM). Typically, an FPKM > 1 indicates that the gene is expressed. Appendix A shows detailed gene expression levels.

#### 3.1.2. Identification and Analysis of DEGs

To determine molecular differences in responses (based on read count) between experimental and control groups, we identified 3377 differentially transcribed (FDR < 0.05, ∣log_2_FC∣ ≥ 1) genes, with 1182 genes (35.00%) upregulated and 2195 (65.00%) genes downregulated (Figure 1).

Functional assignments were defined using GO terms (http://www.genontology.org/, accessed on 5 September 2018), which provide a comprehensive functional classification of genes and their products based on various biological processes (BP), cellular components (CC), and molecular functions (MF). GO functional enrichment analysis revealed several enriched terms: 13 MF terms, mainly involved in oxidoreductase activity, catalytic activity, structural constituents of ribosomes, and monooxygenase activity; 18 CC terms, including intracellular organelle parts, intracellular parts, cell parts, and macromolecular complexes; and 65 BP terms, including a cellular component organisation or biogenesis, oxidation-reduction processes, single-organism metabolic processes, single-organism processes, and cellular protein metabolic processes (Figure 2 and Appendix A).

Pathway-based analysis was performed using the KEGG (http://www.genome.ad.jp/kegg/, accessed on 1 November 2023) pathway database to explore the biological functions and interactions in the genes. The results showed that 3377 DEGs were involved in 110 pathways. Appendix A shows the top ten up- and downregulated genes enriched in KEGG. For the downregulated genes, the most enriched pathways were those involved in aflatoxin biosynthesis, followed by starch and sucrose metabolism and tyrosine metabolism. This suggests that the expression of genes involved in aflatoxin synthesis and the glucose metabolic pathway was suppressed in the experimental group. In contrast, the most enriched pathways in the upregulated genes were valine, leucine, and isoleucine biosynthesis, followed by alanine, aspartate, and glutamate metabolism. This indicates that genes involved in amino acid metabolism were highly expressed in the experimental groups. Furthermore, KEGG metabolic pathway analysis confirmed that the enrichment of valine, leucine, isoleucine, and aflatoxin biosynthesis was significant at *p* ≤ 0.05 (Figure 3).

To investigate the effect of STC on secondary metabolism synthesis in *A. flavus*, we used data from the Center for Integrated Fungal Research (http://weir.statgen.ncsu.edu/aspergillus/chromosomes.php, accessed on 19 March 2019) and employed SMURF (version 3.3.2) (http://www.jcvi.org/smurf, accessed on 19 March 2019) software to analyse 55 secondary metabolic gene clusters from *A. flavus*. Our analysis revealed that the expression of most gene clusters remained unaffected by STC. Further comparative analysis of secondary metabolism gene expression showed that among the 639 secondary metabolism examined genes, 162 genes exhibited significant differences in expression (q-value ≤ 0.001). Notably, 17 of these genes were classified as backbone genes; AFLA_004450, AFLA_010020, AFLA_064560, AFLA_069330, and AFLA_139490 encoded mainly non-ribosomal peptide synthetase (NRPS); AFLA_006170, AFLA_10545, AFLA_128060, AFLA_137870, and AFLA_139410 were responsible for polyketide synthase, PKS. Furthermore, AFLA_017840, AFLA_023020, AFLA_079400, and AFLA_105190 encoded NRPS-like synthase whereas AFLA_028660, AFLA_066890, and AFLA_139480 encoded mitochondrial translation initiation factor IF-2, cytochrome P450, and hybrid PKS/NRPS enzymes, respectively.

Most secondary metabolic gene clusters were downregulated, while only a few were upregulated. Specifically, within the gene cluster 2^#^, the backbone gene AFLA_004450, which encodes dimethylallyl transferase, was upregulated, while AFLA_004370 and AFLA_004400 were downregulated. Similarly, in the 11^#^ gene cluster, only AFLA_028660 was upregulated, while the other genes in this cluster did not exhibit significant differences in expression. Furthermore, within the 27^#^ gene cluster, the genes AFLA_082220, AFLA_082230, and AFLA_082250, and AFLA_101690 and AFLA_101710 in the 33^#^ gene cluster were all upregulated to varying degrees. Among the upregulated genes, AFLA_112890 and AFLA_066740 encoded major facilitator superfamily transporters (MFS), which play important roles in the transfer of some metabolites. Furthermore, genes involved in amino acid synthesis were differentially expressed to varying degrees. For example, AFLA_005510, AFLA_041550, AFLA_083270, and AFLA_126970, which encode amino acid permease, cysteine-lyase, GABA permease, and arginine permease, respectively, were upregulated, while AFLA_038620 (encoding branched amino acid transferase) and AFLA_062910 (encoding specific proline permease) were downregulated.

AFB_1_ synthesis genes were located in the 54^#^ gene cluster. In this study, 30 genes were downregulated to varying degrees. Notably, *norB* was completely repressed in the experimental group (Table 2). However, there was no significant difference in the expression of the global regulatory genes *laeA* and *veA* for secondary metabolites. Additionally, the gene *brlA*, which is involved in growth, was downregulated.

### 3.2. Proteomic Analysis of A. flavus Cultured in PDB with STC

#### 3.2.1. Protein Concentration Protection in *A. flavus*

Appendix A shows the protein concentrations of the samples. As shown in the table, the protein concentration of each sample was >4 mg/mL, and the total amount of proteins was >0.7 mg, indicating that the protein extraction was successful, meeting the requirements for identification and quantification based on purity and total amount. The SDS-PAGE electrophoresis diagram (Figure 4) shows clear and well-separated protein bands without trailing or dispersion. This indicates that high-quality proteins were extracted from *A. flavus*.

#### 3.2.2. Differential Protein Analysis

Overall, 331 differential proteins (FC > 1.5, *p* < 0.05) were identified, comprising 48 upregulated proteins and 283 downregulated proteins (Figure 5). To gain insights into the function of the differential proteins and the differences between the samples at the functional level, we performed GO functional annotation for the differential proteins of CK (CKPs) and TJ (TJPs) (Figure 6). The figure shows that the different proteins in both groups were involved in metabolic processes, detoxification, multi-organism processes, cell component organisation or biogenesis, response to stimulus, localisation, and biological regulation. Notably, proteins involved in cellular and metabolic processes accounted for 22.4% and 29.3% of the total number of proteins, respectively. The number of TJPs in the metabolic process was observed to be higher than that in CKPs, indicating a decrease in the metabolic capacity of cells in the additive group. In addition, proteins involved in detoxification and multiple organism processes were exclusively expressed in the additive group. In the cell component classification, the regions with abundant expression of DEPs included the virion part, membrane-enclosed lumen, organelle part, protein-containing complex, and extracellular region. Organelles, membranes, and cell parts had the highest number of proteins. Similarly, three proteins were exclusively expressed in the addition group: the virion, supramellar, and extracellular regions. Molecular function analysis, as shown in the green parts of Figure 6, successfully identified the GO annotation of the DEPs through mass spectrometry. The activities included structural module, transcription regulator, and transporter activities. Notably, antioxidant activity, binding, and catalytic activity were the predominant molecular functions among the different proteins in *A. flavus*, with the highest numbers of binding and catalytically active proteins. In addition, 26.6% and 59.9% of the proteins expressed in the addition group exhibited structural activity.

A comprehensive analysis was conducted at the functional level to elucidate the functional enrichment of the differentially expressed proteins. Significant enrichment analysis of GO function was performed on the identified DEPs (Figure 7). Each column in the figure represents a specific GO function. The horizontal coordinates represent GO, and classification names and the vertical coordinates represent the enrichment rate. The darker the colour, the more pronounced the enrichment. Antioxidant activity-related proteins and protein-containing proteins were mainly associated with cell parts, organelle parts, cellular processes, and structural molecule activity-related proteins activity. GO analysis showed that the number of proteins involved in catalytic activity in the additive group was the highest at 185.

To gain further insights into the function of DEPs in metabolic pathways, we conducted an analysis of the biological pathways associated with the proteins identified in the addition and blank groups. As previously mentioned, 48 upregulated and 283 downregulated proteins were successfully identified in this study. Through KEGG annotation, 67 KEGG pathways were identified (Appendix A). The pathways associated with these differential proteins were mainly concentrated in metabolism (56), environmental information processing (1), genetic information processing (7), and cellular processes (3). No differential proteins were found in other categories.

In this study, we considered a pathway with a *p* < 0.05 significantly enriched with differential proteins. After analysis, 19 pathways were identified, and the top 10 pathways with the most significant enrichment were the aflatoxin synthesis pathway, glycolysis/gluconeogenesis, glutathione metabolism, carbon metabolism, glyoxylic acid and dicarboxylic acid metabolism, tryptophan metabolism, starch, and sucrose metabolism, methane metabolism, fructose and mannose metabolism, and antibiotic synthesis. Based on the *p*-values, the aflatoxin synthesis pathway, glycolysis/gluconeogenesis, and glutathione metabolism were assumed to play major roles.

Table 3 lists the top five enriched metabolic pathways in the downregulated proteins. These pathways include aflatoxin biosynthesis (aflatoxin biosynthesis), glutathione metabolism (glutathione metabolism), tryptophan metabolism (tryptophan metabolism), starch and sucrose metabolism (starch and sucrose metabolism), fructose, and mannose metabolism. Starch and sucrose metabolism, and fructose and maltose metabolism are involved in energy metabolism, and proteins related to glucose metabolism are downregulated, including glycogen synthetase (AFLA_004660), α-1.4-glucosidase (AFLA_122400), β-1.3-glucosyltransferase (AFLA_068300), and 1.6-fructose diphosphatase (AFLA_073670). Additionally, certain proteins involved in the glutathione metabolic pathway, such as glutathione transferase (AFLA_016400 and AFLA_046620) and glucose-6-phosphate dehydrogenase (AFLA_086620), were downregulated. In the tryptophan metabolic pathway, acetaldehyde dehydrogenase (AFLA_10879) and hyphal catalase (AFLA_090690) were also downregulated.

To investigate aflatoxin biosynthesis-related proteins, the MS data were searched, and 12 proteins matching the metabolic pathway of *A. flavus* were identified. Among them, AflP (AFLA_139210), AflO (AFLA_139220), AflK (AFLA_139110), AflJ (AFLA_139320), AflC (AFLA_139410), AflG (AFLA_139260), and AflH (AFLA_1) 39330) proteins were downregulated (Table 3). The initial FAS α and β subunits of aflatoxin synthesis (AFLA_046360 and AFLA_139410), NOR reductase (AFLA_139390 and AFLA_112820), and aflQ redotase (AFLA_139200) were not detected in the addition group. AflQ was responsible for converting OMST to AFB1 in the *A. flavus* synthetic pathway, indicating that STC had no effect on this catalytic process.

## 4. Discussion

Numerous studies have focused on controlling aflatoxin production, and some natural substances, such as onion, garlic, eugenol, khellin, caffeine, piperlongumine, luteolin, saccharol, resveratrol, and tannic acid, have been identified as inhibitors of aflatoxin synthesis [36]. However, these are all exogenous substances. STC served as a precursor for aflatoxin synthesis by being spiked in the medium with *A. flavus*. Transcriptome analysis revealed that 30 genes in the AFB_1_ biosynthetic gene cluster were downregulated to varying degrees. Furthermore, KEGG analysis showed that branched-chain amino acids (BCAAs) were enriched in most DEGs. The proteomic analysis also demonstrated that 12 DEPs matched the metabolic pathway of *A. flavus*.

Secondary metabolic pathways in fungi are relatively complex and involve various enzymes, such as polyketone synthase, epoxide hydrolase, methyl transferase, reductase, dehydrogenase, cytochrome P450 monooxygenase, and fatty acid synthase. However, a single type of enzyme can be produced by multiple genes, making it challenging to determine the involvement of a specific gene in the secondary metabolism. Using metabolic analysis tools and SMURF software, Georgianna et al. identified 55 secondary metabolic pathways in *A. flavus*, including 27 NRPS, 22 PKS, and metabolic pathways for ciprofloxacin synthesis, spore pigment synthesis, and aflatoxin synthesis, each serving distinct biological functions [37]. *AflR* is a transcriptional regulatory gene crucial for aflatoxin synthesis, as it activates almost all the structural genes involved in the process. *AflS* acts as an auxiliary transcription factor, cooperating with *aflR* to prevent inhibition and ensure aflatoxin synthesis. In this study, we found no significant difference in the expression of LaeA and veA, which are global regulatory factors involved in aflatoxin synthesis.

Aflatoxins are highly oxidative metabolic products produced during oxidative stress [38]. Our transcriptome analysis revealed an increase in Cu–Zn superoxide dismutase expression, which is consistent with findings in previous research [36]. A total of 30 genes involved in aflatoxin synthesis were downregulated, demonstrating that STC directly inhibits the genes responsible for encoding enzymes within this gene cluster, consequently inhibiting aflatoxin production. Furthermore, certain genes that encoded antioxidant enzymes that were upregulated can also inhibit aflatoxin synthesis in the presence of STC.

The role of amino acid metabolism in aflatoxin synthesis is complex. Some amino acids can serve as carbon and nitrogen sources for growth and synthesis of aflatoxin. In the case of *A. flavus*, phenylalanine, tyrosine, tryptophan, proline, and arginine are used for aflatoxin synthesis [39]. Arginine is particularly crucial for aflatoxin synthesis in *A. parasiticus* and can substitute aspartate and alanine in the process [40,41]. KEGG enrichment analysis revealed that among the 67 metabolic pathways, BCAA metabolic activity was associated with aflatoxin biosynthesis, while the metabolism of other basic amino acids and acidic amino acids was mainly related to fungal growth. Furthermore, 18 ribosomal biosynthetic genes were upregulated, suggesting that STC may promote the growth of *A. flavus* to a certain extent but inhibit the synthesis of aflatoxins. These findings are consistent with those of previous studies [42].

STC serves as a precursor for aflatoxin synthesis. Theoretically, an increase in STC levels should enhance the ability of isolates to synthesise aflatoxins in these strains that produce aflatoxins. However, cellular and secondary microbial metabolisms are complex processes. Further, while precursor substances can inhibit metabolic production, this phenomenon has not been previously reported for aflatoxin synthesis, despite extensive research on antibiotic synthesis. Generally, adding exogenous precursors during antibiotic synthesis can control the direction of synthesis and increase antibiotic production. For example, adding precursors inhibits the synthesis of epothilone, a secondary polyketide metabolite [43]. Similarly, FK506 production was significantly improved during fermentation in the presence of precursors, albeit at low concentrations [44]. In the case of *A. flavus* CGMCC, aflatoxin synthesis decreased in the presence of STC. However, under the same conditions, aflatoxin production increased in other isolates. This confirms that aflatoxin synthesis is isolate-specific and can vary depending on the nutrient medium, external environment, and degree of tolerance to particular substances [45].

The synthesis of secondary metabolites is closely related to primary metabolism because cellular energy, precursors, and co-factors can potentially limit secondary metabolism. Many enzymes involved in aflatoxin biosynthesis and catalytic reactions rely on NADPH. Glucose can control aflatoxin biosynthesis via NADPH, consequently repressing carbon metabolism and the tricarboxylic acid cycle. Therefore, the cellular balance of NADPH/NADP^+^ may affect aflatoxin synthesis [46,47]. Lipid metabolism is also associated with aflatoxin synthesis. The NADPH/NADP ratio determines whether acetyl-CoA enters fatty acid metabolism or aflatoxin anabolism. A high proportion of NADPH/NADP may enable acetyl-CoA to enter fatty acid metabolic pathways, while low levels may encourage its entry into other pathways [48]. The pentose phosphate pathway is the main source of intracellular NADPH in organisms, and it is possible that certain genes with significant differences may play a role as additional factors influencing aflatoxin synthesis.

In this study, iTRAQ was used to determine the intracellular content of *A. flavus* in media with and without STC. Overall, 331 DEPs were identified using mass spectrometry. GO analysis revealed that the additive group proteins were more involved in metabolic pathways than those in the control group. Notably, most of the enzymes associated with glucose metabolism were downregulated, such as those involved in starch, sucrose, fructose, and maltose metabolism. It is well known that sugar is the main energy source for living organisms, with a significant amount of energy released through oxidation to meet the needs of life activities. The downregulation of α-amylase (AFLA_122400) and glucose transfer enzyme (AFLA_068300) indicated that *A. flavus* has a reduced ability to hydrolyse starch into glucose, which is conducive to forming aflatoxins. Woloshuk et al. reported the relationship between α-amylase activity and aflatoxin production [49]. In addition, the expression of glycogen branching enzyme AFLA_081340 and glycogen debranching enzyme AFLA_119670 was downregulated, indicating the regulation of glycogen synthesis. This finding is consistent with the downregulation of glycogen synthase (AFLA_004660) expression. During the metabolism of fructose and maltose, the activities of fructose 1, 6-diphosphatase (AFLA_027310 and AFLA_073670) decreased. This enzyme plays a crucial role in the gluconeogenic metabolic pathway within living organisms and is present in the cytoplasm.

However, the proteins associated with carbon and glycolytic/gluconeogenic metabolism were also upregulated. Glycolysis is the enzymatic degradation of glucose into pyruvate, accompanied by ATP production. The pyruvate generated from glycolysis enters the mitochondria and undergoes catalysis by pyruvate dehydrogenase to convert it into acetyl CoA, which serves as the link between glycolysis and the tricarboxylic acid cycle. In this study, pyruvate decarboxylase (AFLA_031570) in the pyruvate dehydrogenase complex was significantly upregulated (*p* = 0.003), indicating that pyruvate in the additive group was effectively converted to acetyl-CoA. Simultaneously, the activities of other related enzymes in the tricarboxylic acid cycle were also upregulated. The carbohydrate metabolism of *A. flavus* in the additive group was active, facilitating the maintenance of normal functions, which is consistent with the transcriptome findings in this study.

Amino acid metabolism, arising from protein hydrolysis, is an indispensable metabolic pathway. Metabolism of various amino acids in the body includes two processes., First, they are used for the synthesis of proteins, peptides, and other nitrogen-containing substances for the body’s needs. Second, they are decomposed into alpha-ketoacids, amines, and carbon dioxide through processes, such as transamination, deamination, decarboxylation, and removal in combination with ammonia or decarboxylation. The resulting α-ketoacid obtained from decomposition can be converted into certain non-essential amino acids, and it can be converted into sugars and lipids. In this study, differential protein analysis between the additive and control groups revealed that tryptophan metabolic pathway protein expression was downregulated, while that of glycine, serine, and threonine was upregulated. These findings suggest that the two amino acids are involved in different metabolic pathways in *A. flavus*.

In addition, proteomic analysis revealed that the expression of some enzymes involved in glutathione metabolism, especially glutathione transferase (AFLA_016400, AFLA_046620, AFLA_119330, and AFLA_031820) was downregulated. Saxena et al. found that the activity of glycine transferase in *A. flavus* is positively correlated with aflatoxin metabolism [50]. Similarly, Allameh et al. observed significantly higher glutathione transferase activity in virus-producing strains than in non-virus-producing strains of *A. parasiticus* [51]. Therefore, decreased glutathione transferase activity may inhibit aflatoxin synthesis.

## 5. Conclusions

In this study, we identified 3377 differentially expressed genes, with 1182 genes upregulated and 2195 genes downregulated. STC had the least influence on 55 secondary metabolic gene clusters; however, 30 genes were expressed to varying degrees within the aflatoxin synthesis gene cluster. Notably, the suppression of *norB* in the experimental group suggests that STC probably increased oxidoreduction enzyme activity, facilitated the growth of isolates and improved the branched-chain amino acid biosynthesis, and subsequently inhibited the synthesis of aflatoxins. Overall, 331 DEPs were identified in *A. flavus*, among which 48 were upregulated and 283 downregulated. GO and KEGG enrichment analysis of these proteins highlighted the upregulation of glucose-related enzyme activity and downregulation of glutathione transferase activity, which may be the primary factors contributing to the inhibition of synthesis. Further, within the aflatoxin synthesis pathway, twelve proteins were identified, seven of which were downregulated. AflG exhibited the largest difference in expression, which may be the key enzyme mediating STC inhibition of aflatoxin synthesis. This study provides the first report on the response of *A. flavus* to STC. These findings provide valuable insights that may serve as a reference for further studies on the effects of aflatoxin synthesis intermediates on the synthesis of aflatoxin which is crucial for effectively inhibiting aflatoxin contamination in food.

## Figures and Tables

**Figure 1 jof-09-01193-f001:**
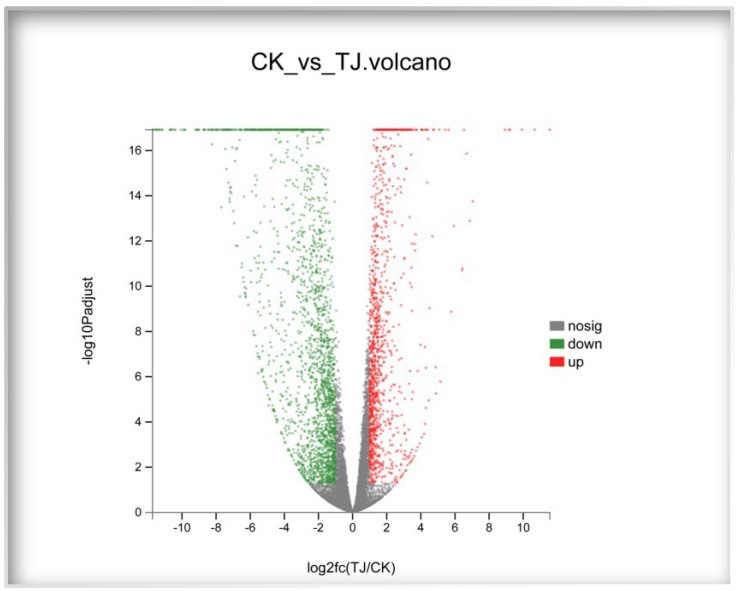
Volcano map of the differentially expressed gene.

**Figure 2 jof-09-01193-f002:**
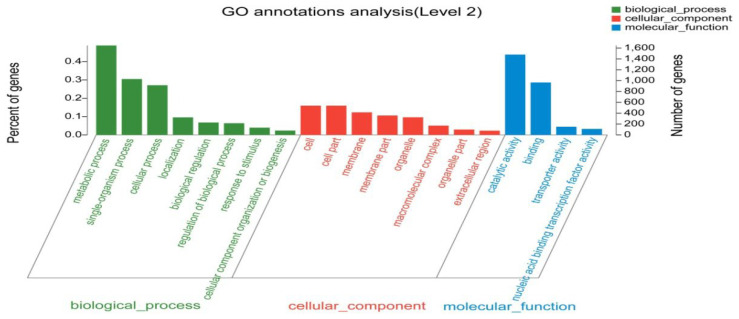
Gene ontology analysis of differently expressed genes.

**Figure 3 jof-09-01193-f003:**
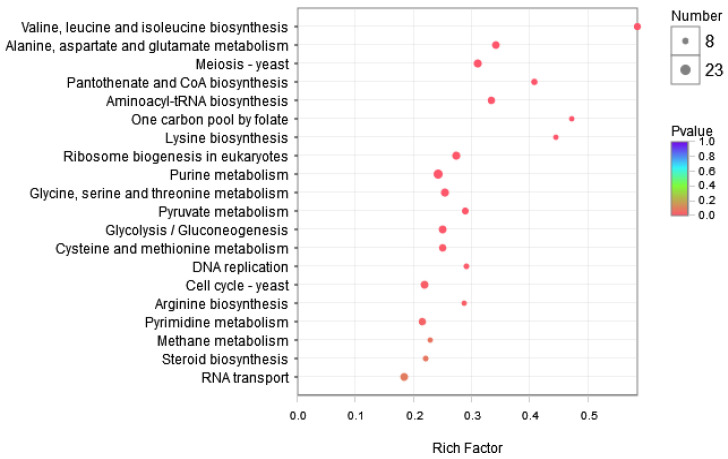
Bubble maps of the top 10 KEGG pathways of downregulated and upregulated genes.

**Figure 4 jof-09-01193-f004:**
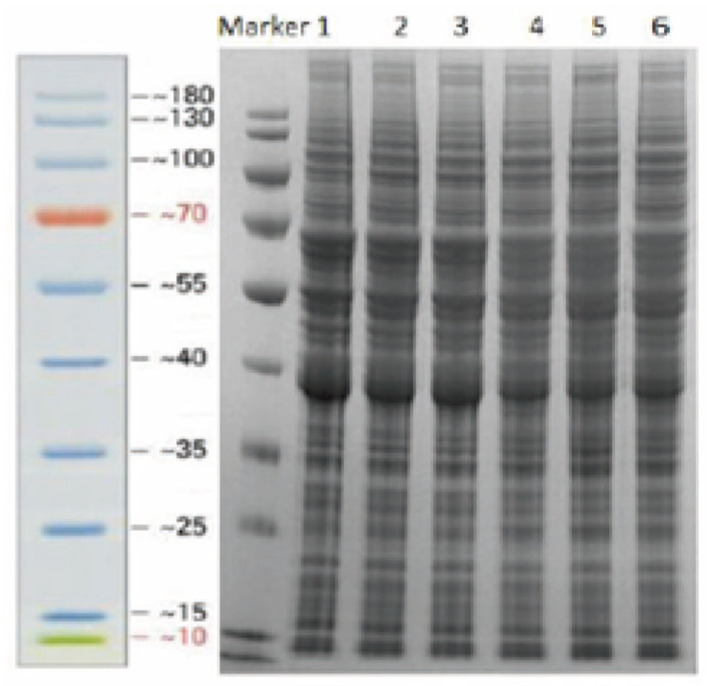
SDS-PAGE of six samples.

**Figure 5 jof-09-01193-f005:**
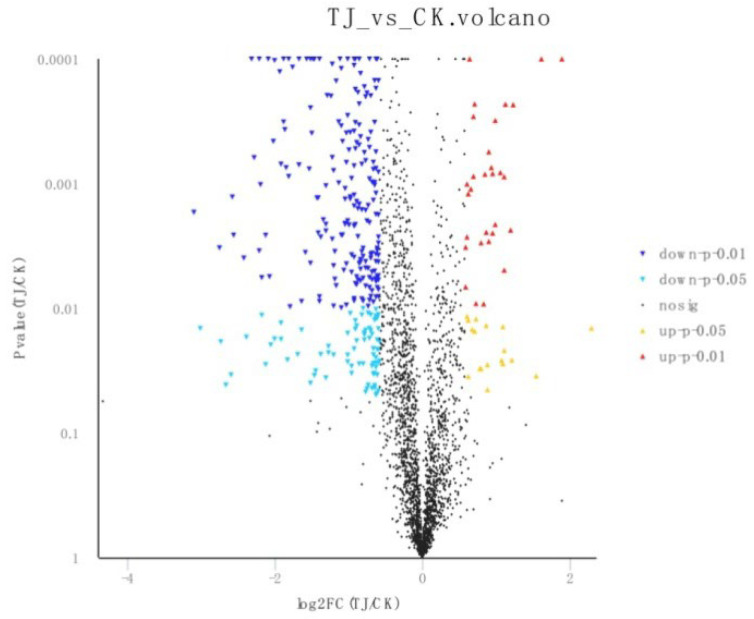
Relative abundances of differentially expressed quantitative protein in *Aspergillus flavus*. (X: protein ratios taken in base 2 logarithmic; Y: *p*-value).

**Figure 6 jof-09-01193-f006:**
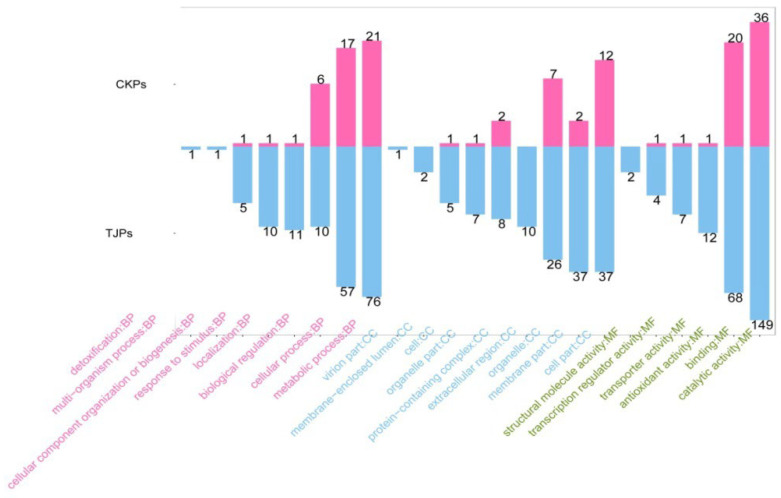
The GO annotation of different expressed protein.

**Figure 7 jof-09-01193-f007:**
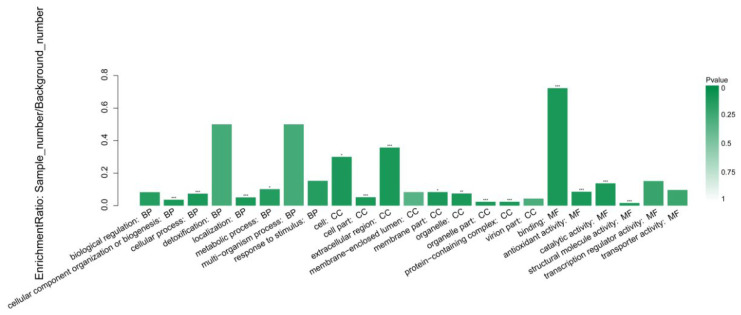
GO enrichment analysis of different expressed proteins. *: FDR < 0.05; **: FDR < 0.01; ***: FDR < 0.001.

**Table 1 jof-09-01193-t001:** Search parameters of Proteome Discoverer.

Item	Value
Proteome Discoverer version	2.1
Protein Database	*Aspergillus_flavus*.JCVI-afl1-v2.0.pep.all.fasta
Cys alkylation	Iodoacetamide
Dynamic Modification	Oxidation (M), Acetyl (Protein N-Terminus), iTRAQ8 plex (Y)
Static Modification	iTRAQ8 plex (K), iTRAQ8 plex (N-Terminus), Carbamidomethyl (C)
Enzyme Name	Trypsin (Full)
Max. Missed Cleavage Sites	2
Precursor Mass Tolorance	20 ppm
Fragment Mass Tolorance	0.1 Da
Validation based on	q-value

Note: False discovery rate (FDR) of peptide identification was set as ≤0.01.

**Table 2 jof-09-01193-t002:** Statistics of differentially expressed genes in aflatoxin synthesis.

Gene_Id	Expression Values (RPKM)		Log_2_FC (TJ/CK)	Gene Function Description	Expression Regulation
	**CK**	**TJ**			
AFLA_139200	152.1	0.089	−9.792	aflQ/ordA/ord-1/oxidoreductase/cytochrome P450 monooxigenase	down
AFLA_139330	3153.503	1.854	−9.854	aflH/adhA/Short-chain alcohol dehydrogenase	down
AFLA_125300	8.129	1.8	−1.96	aflD/nor-1/Norsolorinic acid ketoreductase	down
AFLA_139240	1044.919	0.238	−10.69	aflLa/hypB/hypothetical protein	down
AFLA_066940	8.593	0.134	−5.037	aflO/omtB/dmtA/O-methyltransferase	down
AFLA_139320	1900.32	0.484	−11.16	aflJ/estA/esterase	down
AFLA_139410	196.808	4.375	−5.187	aflC/pksA/pksL1/polyketide synthase	down
AFLA_139190	728.711	0.115	−11.73	aflK/vbs/VERB synthase	down
AFLA_139220	2925.138	0.576	−11.66	aflO/omtB/dmtA/O-methyltransferase	down
AFLA_139400	2940.438	1.016	−10.62	aflCa/hypC/hypothetical protein	down
AFLA_139390	2307.841	5.638	−7.266	aflD/nor-1/reductase	down
AFLA_139260	786.053	0.138	−11.47	aflG/avnA/ord-1/cytochrome P450 monooxygenase	down
AFLA_139210	917.681	0.221	−11.29	aflP/omtA/omt-1/O-methyltransferase	down
AFLA_046360	27.569	99.777	2.076	acetyl-CoA carboxylase	up

**Table 3 jof-09-01193-t003:** Top 5 KEGG pathways enriched in downregulated proteins.

Gene	Product
**Aflatoxin biosynthesis**	
AFLA_139210	aflP/omtA/omt-1/O-methyltransferase A
AFLA_139220	aflO/omtB/dmtA/O-methyltransferase B
AFLA_139190	aflK/vbs/VERB synthase
AFLA_139320	aflJ/estA/esterase
AFLA_139410	aflC/pksA/pksL1/polyketide synthase
AFLA_139260	aflG/avnA/ord-1/cytochrome P450 monooxygenase
AFLA_139330	aflH/adhA/short-chain alcohol dehydrogenase
**Glutathione metabolism**	
AFLA_010790	elongation factor 1 gamma, putative
AFLA_010930	glutathione S-transferase family protein, putative
AFLA_016400	glutathione-S-transferase, putative
AFLA_086620	glucose-6-phosphate 1-dehydrogenase
AFLA_079910	glutathione peroxidase Hyr1, putative
AFLA_046620	glutathione-S-transferase, putative
AFLA_119330	glutathione-S-transferase theta, GST, putative
AFLA_031820	glutathione-S-transferase GstA
AFLA_083370	glutathione oxidoreductase Glr1, putative
**Tryptophan metabolism**	
AFLA_10879	aldehyde dehydrogenase AldA, putative
0AFLA_034380	catalase, putative
AFLA_090690	mycelial catalase Cat1
AFLA_050600	betaine-aldehyde dehydrogenase, putative
AFLA_056170	spore-specific catalase CatA
AFLA_122110	bifunctional catalase-peroxidase Cat2
**Starch and sucrose metabolism**	
AFLA_004660	glycogen synthase Gsy1, putative
AFLA_122400	glucan 1,4-alpha-glucosidase, putative
AFLA_018550	glycogen phosphorylase GlpV/Gph1, putative
AFLA_068300	1,3-beta-glucanosyltransferase Bgt1
AFLA_081340	glycogen debranching enzyme Gdb1, putative
AFLA_119670	glycogen branching enzyme GbeA, putative
**Fructose and mannose metabolism**	
AFLA_015580	sorbitol/xylulose reductase Sou1-like, putative
AFLA_098700	carbonyl reductase, putative
AFLA_133940	conserved hypothetical protein
AFLA_027310	fructose-1,6-bisphosphatase Fbp1, putative
AFLA_073670	fructose-1,6-bisphosphatase Fbp1, putative

## Data Availability

The data presented in this study are included in this article and its Appendix A. The raw RNA-seq data generated in this study are openly available in the Sequence Read Archive (SRA), accession number PRJNA759939.

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
