# Peer review of "Transcriptomic and Proteomic Insights into the Effect of Sterigmatocystin on Aspergillus flavus"

_jof, 2023, doi:10.3390/jof9121193_

Round 1
Reviewer 1 Report
Comments and Suggestions for Authors
The Manuscript under the title “Transcriptomic and Proteomic Analysis of Interaction Between Aflatoxin B1 and Sterigmatocystin” aimed to investigate differentially expressed genes (DEGs) and differentially expressed proteins (DEPs) in A. flavus after introducing STC into the matrix using a combination of transcriptomic and proteomic data to gain new insight into the control of A. flavus and the mechanism that regulates mycotoxin production.
Considering the importance of the control of A. flavus and the mechanisms that regulate mycotoxin production and their potent carcinogenic effect on humans and animals, this study is relevant and the article may be published in its present form.
Reviewer 2 Report
Comments and Suggestions for Authors
The manuscript describes proteomic and transcriptomic analysis of the effect of STC on A. flavus. The experiments were properly designed and carried out. The results are valuable and deserve publishing. Evaluation, interpretation, and discussion of the results, however, should be improved. Furthermore, the title is inadequate (the preprint had a better title), and the contradiction regarding the use of STC as a biomarker for AFB should be clarified.
Title : the manuscript does not describe interaction between AFB1 and STC! It is about the effect of STC on transcriptome and proteome of a fungus that produces AFB1. The puzzling statement is repeated in the Abstract: "AFB1 interacts with another mycotoxin sterigmatocystin (STC), which is also a precursor of AFB1..." How does AFB1 "interact" with its precursor STC? I am not aware of any interaction, and the authors do not support their claim with any reference or experimental data. They used a better title in their preprint "Transcriptome Insights into the Effect of Sterigmatocystin on Growth and Alatoxin B1 Production by Aspergillus Flavus". A good and precise title could be, e.g. "Transcriptome and Proteome Insights into the Effect of Sterigmatocystin on Aspergillus Flavus".
The Introduction contains grammatically correct yet imprecise sentences:
L49: "STC shares a biosynthetic pathway with AFB1": Sharing a pathway normally means that two metabolites have a same precursor, after which the pathways forks into two branches, each leading to one metabolite. The relationship between STC and AFB is different, as STC is a precursor of AFB.
57: "(De Saeger and personal communication, 2012)": No need to support an established fact with "personal communication", citing [17-20] suffices. Furthermore, a sufficiently sensitive analytical method will detect STC in any food sample contaminated with AFB.
RESULTS
The transcriptomics was well done, but evaluation and presentation is suboptimal. Estimating statistical significance in genome-wide transcriptomics with "Bonferroni-corrected P-value ≤ 0.05", as stated in Methods, is inadequate. edgeR-package, which was used for the analysis, offers several standard and advanced methods, why was none of them used. Or an adequate method such as FDR was actually used but the description of the statistics in Methods was erroneous?
Interpretation: The 3377 differentially expressed genes lump many marginally affected genes and noise with few strongly affected genes. This can also be seen on the Volcano plot, which is hardly readable (please provide a better resolution). A list of most strongly affected pathways and genes should be provided, similarly as in proteomics (Table 3). Optimally, the pathways affected on the level of proteome and transcriptome could be shown side by side in a common table, but two similar tables would also be OK. Adding a quantitative measure of the effect, e.g. changefold factor, would be helpful.
Proteomics: The low number of up- and downregulated proteins contrast with the high number of affected genes. This discrepancy has be addressed. I suppose that the reason was a low number of protein-coded gene to which peptides were assigned; this number should be reported. A low coverage would explain the discrepancy; if the coverage of the proteome was good, the discrepancy is striking, and it has to be addressed in the discussion.
414 – 415: "STC serves as a precursor for aflatoxin synthesis. Theoretically, an increase in STC levels should enhance the ability of isolates to synthesise aflatoxins." This passage is well meant but misleading: STC produced by the fungus will accumulate to high levels in strains with the synthesis blocked after aflO, but these strains will produce no AFB. If all genes after aflO are active, STC will be low, because it was used up by aflP, but AFB will accumulate to high levels. This is also relevant for the following:
L67-70: the authors write about the use of STC as "potentially warning markers for aflatoxin contamination", which contradicts their statement on L45-47 that many fungi produce STC but not AFB. Accumulation of STC by these fungi would lead to false positive prediction of AFB contamination. The authors themselves question the idea of using STC as a biomarker for AFB, when they write:
52-55 "Therefore, food infected with these fungi often have high STC concentrations. In contrast,/ when A. flavus and A. parasiticus, which are capable of metabolising STC, invade food, STC concentrations are generally low." This contradiction should be addressed.
415-416: "However, cellular and secondary microbial metabolisms are complex processes." It is a trivial fact that metabolism (no plural!) is complex. How is it relevant or useful for the topic discussed here?
Reviewer 3 Report
Comments and Suggestions for Authors
In this study, transcriptomic and proteomic data were used to investigate the differentially expressed genes (DEGs) and differentially expressed proteins (DEPs) of A. flavus after the introduction of STC into the substrate. The results of this study provide a new understanding of the control of A. flavus and the regulatory mechanism of mycotoxin production, and have important significance for the prevention and control of food poisoning, after review, I suggest that this manuscript can be published in this journal after minor revisions.
1. Line 53-55: You are advised to check the spelling carefully. For example, there are redundant “/” in “In contrast,/ when A. flavus and A. parasiticus, which are capable of metabolising STC, invade food, STC concentrations are generally low”.
2. Table 1: It is recommended to check the full text for non-English characters, spaces and other writing details, for example, “Oxidation (M), Acetyl (Protein N-Terminus) ,iTRAQ8plex (Y)” in Table 1.
3. Line 211and 224: Please standardize full-text illustrations and its title of the article, rather than directly applying the derived figure, especially with “CK-vs-TJ.volcano” in Figure 1, and “GO annotation analysis(Level 2)” in Figure 2.
4. Line 187-372: It is suggested that the first sentence of each paragraph can directly summarize the whole paragraph center in “3. Result”.
5. Line 289 and 319: It is suggested to refer to the illustration of the journal and optimize the figures and tables in the paper one by one. For example, looking at Figure 4 and Figure 6 alone cannot fully understand the content expressed, and corresponding remarks should be added so that it can be independently understood.
6. Line 374-488: It is recommended to increase the transcriptomic and proteomic analysis and discussion of Aflatoxin B1 interaction with sterigmatocystin in a methodical and in-depth way, rather than simply elaborate the phenotypic changes such as up-regulation and down-regulation.
Comments on the Quality of English LanguageQuality of english language is well, but need to be checked out.
